# BENCHMARKING MULTI-AGENT DEEP REINFORCEMENT LEARNING ALGORITHMS

## ABSTRACT

We benchmark commonly used multi-agent deep reinforcement learning (MARL) algorithms on a variety of cooperative multi-agent games. While there has been significant innovation in MARL algorithms, algorithms tend to be tested and tuned on a single domain and their average performance across multiple domains is less characterized. Furthermore, since the hyperparameters of the algorithms are carefully tuned to the task of interest, it is unclear whether hyperparameters can easily be found that allow the algorithm to be repurposed for other cooperative tasks with different reward structure and environment dynamics. To investigate the consistency of the performance of MARL algorithms, we build an open-source library of multi-agent algorithms including DDPG/TD3/SAC with centralized Q functions, PPO with centralized value functions, as well as QMix, and test them across a range of tasks that vary in coordination difficulty and agent number. The domains include the Multi-agent Particle World environments, StarCraftII micromanagement challenges, the Hanabi challenges, and the Hide-And-Seek environments. Finally, we investigate the ease of hyperparameters tuning for each of the algorithms by tuning hyperparameters in one environment per domain and re-using them in the other environments within the domain. The open-source code and more details can be found in our website: https://sites.google.com/view/marlbenchmarks.

## 1 INTRODUCTION

Widespread availability of high-speed computing, neural network architectures, and advances in reinforcement learning (RL) algorithms have led to a continuing series of interesting results in building cooperative artificial agents: agents collectively playing Hanabi to an expert level (Hu & Foerster, 2019), designing cooperative StarCraftII bots (Rashid et al., 2018) that outperform hand-designed heuristics, and constructing emergent languages between agents (Mordatch & Abbeel, 2017). Each of these aforementioned results have often come with the introduction of a new algorithm, leading to a proliferation of new algorithms that is rapidly advancing the field.

However, these algorithms are often designed and tuned to get optimal performance in a particular deployment environment. In particular, it is not unusual for each new algorithm to come with a new proposed benchmark on which it is evaluated. Consequently it is not obvious that these algorithms can easily be re-purposed for new tasks; subtle interactions between the algorithm, the architecture and the environment may lead to high asymptotic performance on one task and total failure when applied to a new task. Without examining an algorithm across a range of tasks, it is difficult to assess how general purpose it is.

Furthermore, the high asymptotic rewards that are often presented may hide complexities in using the algorithms in practice. The amount of time that researchers spent in finding optimal hyperparameters is often obscured, making it unclear how extensive of a hyperparameter search was needed to find the good hyperparameters. That is, researchers will often report a grid search of hyperparameters but not the prior work that was done to pick out a hyperparameter grid that actually contained good hyperparameters. Furthermore, the amount of computation provided to tune the studied algorithm may not be provided to the baseline algorithms that it will be compared against. This can lead to an inflated performance of the proposed algorithm relative to the benchmarks. All these problems can arise without any ill intent on the part of the authors, but they make the problem of assessing algorithms quite challenging.

The downstream consequence of this proliferation of algorithms coupled with an absence of standard benchmarks is a lack of clarity on the part of practitioners as to which algorithm will give consistent, high performance with minimal tuning. Researchers are often operating under computational constraints that limit how extensive of a hyper-parameter sweep they can perform; the ease with which good hyperparameters can be found is consequently a useful metric. When tackling a new multi-agent problem, researchers have no clear answer to the questions: 1) which MARL algorithm should I use to maximize performance and 2) given my computational resources, which algorithm is likeliest to work under my constraints?

We present an attempt to evaluate the performance, robustness and the relative ease of using these algorithms by benchmarking them across a wide variety of environments that vary in both agent number, exploration difficulty, and coordination complexity. By exploring a large range of possible environments, we identify algorithms that perform well on average and serve as a strong starting point for a variety of problems. We tackle the question of relative difficulty in finding hyperparameters by looking at how hyperparameters transfer across settings: tuning hyperparameters on one set of environments and applying them without re-tuning on the remaining environments. Using this procedure, we can provide effective recommendations on algorithm choice for researchers attempting to deploy deep multi-agent reinforcement learning while operating under constrained hyper-parameter budgets. We test Proximal Policy Optimization (Schulman et al., 2017) with centralized value functions (MAPPO), Multi-Agent DDPG (MADDPG) (Lowe et al., 2017), Multi-Agent TD3 (Fujimoto et al., 2018a) (MATD3), a Multi-Agent variant of Soft Actor Critic (Haarnoja et al., 2018) (MASAC), and QMix (Rashid et al., 2018). We focus specifically on the performance of these algorithms on fully cooperative tasks, as this avoids game theoretic issues around computing the distance to Nash equilibria, and allows us to solely characterize performance in terms of asymptotic reward.

The contributions of this paper are the following

- Benchmarking multi-agent variants of single-agent algorithms across a wide range of possible tasks including StarCraftII micromanagement (Rashid et al., 2019), Multi-agent Particle World (Mordatch & Abbeel, 2017), Hanabi (Bard et al., 2020), and the Hide-And-Seek domain (Baker et al., 2019).

- Establishing that under constrained hyperparameter searching budgets, the multi-agent variant of PPO appears to be the most consistent algorithm across different domains.

- The design and release of a new multi-agent library of various on/off-policy learning algorithms with recurrent policy support.

## 2 RELATED WORK

MARL algorithms have a long history but have, until recently, primarily been applied in tabular settings (Littman, 1994; Busoniu et al., 2008). Notions of using a Q-function that operated on the actions of all agents, known as Joint-Action Learners (Claus & Boutilier, 1998) have existed in the literature since its inception with algorithms like Hyper-Q (Tesauro, 2004) using inferred estimates of other agent strategies in the Q-function. Recent MARL algorithms have built upon these ideas by incorporating neural networks (Tampuu et al., 2017), policy-gradient methods (Foerster et al., 2017), and finding ways to combine local and centralized Q-functions to enable centralized learning with decentralized execution (Lowe et al., 2017; Sunehag et al., 2018).

Alongside the proliferation of algorithms has come a wide variety of new, cooperative MARL benchmarks. Unlike single-agent RL, where MuJoCo (Todorov et al., 2012) and Atari (Mnih et al., 2013) have become standard benchmarks, there is significantly less consensus on appropriate benchmarks. In this work, we consider 4 popular multi-agent environments, which we believe are the most representative in the community. Besides those we considered in this work, other interesting cooperative environments may include MAgent (Zheng et al., 2017), a platform that can efficiently support hundreds of particle agents for cooperative tasks, multi-agent MuJoCo, in which each joint is an independent agent (Schroeder de Witt et al., 2020), and CityFlow (Zhang et al., 2019), which studies large-scale decentralized traffic light control.

There also has been a variety of attempts to benchmark MARL algorithms that differ in scope from our paper. Gupta et al. (2017) benchmarks a similar set of algorithms to ours on a wide variety of environments. However, they do not consider algorithms that train in a centralized fashion

while acting decentralized and instead perform a comparison between fully centralized training and execution and full decentralized algorithms. They establish that parameter sharing is an essential component of getting quick convergence in MARL algorithms. Schroeder de Witt et al. (2020) benchmark algorithms with centralized Q and value functions on a decentralized variant of the MuJoCo environments; however, they primarily study variants of QMix and MADDPG and do not compare with on-policy algorithms.

# 3 MARL ALGORITHMS

## 3.1 PRELIMINARIES

We study decentralized partially observed Markov decision processes (DEC-POMDP) (Oliehoek et al., 2016) with global rewards. A DEC-POMDP is defined by an eight tuple $\langle S, U, P, r, Z, O, n, \gamma \rangle$. $s \in S$ is a state space describing a sufficient set of state variables to make the system dynamics Markovian. For simplicity we assume the agents share action space $U$ and each agent $a \in \{1, \ldots, n\}$ picks an action $u^a \in U$ which are concatenated to form join action $\boldsymbol{u} \in \boldsymbol{U}^n$. We denote the joint action without the action of agent $a$ as $u_{-a}$. Joint actions $\boldsymbol{u}$ and state are fed to state transition function $P(s'|s, \boldsymbol{u}) : S \times U \times S \to [0, 1]$. These are cooperative tasks so all agents share a bounded reward function $r(s, \boldsymbol{u}) : S \times U^n \to \mathcal{R}$ and have shared discount factor $\gamma \in [0, 1]$

Each agent $i$ has an observation function $O^a(s) : S \to Z$ which defines how the global state is mapped onto a local observation $z \in Z$. Each agent maintains an action-observation history $\tau^a \in T \in (Z \times \boldsymbol{U})^*$ which it conditions its policy $\pi^a(u^a|\tau^a) : T \times U \to [0, 1]$ on. Finally, given the joint policy $\pi(\mathbf{u}) = \prod_i \pi^a(u^a|\tau^a)$ we can define a joint value function $V^\pi(s_t) = \mathbb{E}_{s_{t+1:\infty}} \left[ \sum_{i=0}^{\infty} \gamma^i r_{t+i}|s_t \right]$ and joint Q function $Q^\pi(s_t, \mathbf{u}_t) = \mathbb{E}_{s_{t+1:\infty}} \left[ \sum_{i=0}^{\infty} \gamma^i r_{t+i}|s_t, \mathbf{u}_t \right]$.

We assume that the learning algorithm has access to both true states $S$, as well as the trajectories of all agents $\tau^a$. The agents however, only have access to $\tau^a$ for computing their policy. The goal of the agents is to jointly optimize the quantity $J^\pi = \mathbb{E} \left[ \sum_{i=0}^{\infty} \gamma^i r(s_t, \mathbf{u}) \right]$.

## 3.2 BASELINE ALGORITHMS

We introduce all the baseline algorithms we consider, including MADDPG, MATD3, MASAC, QMix and MAPPO. For all problems considered, the action space is discrete. More algorithmic details and the complete pseudo-code can be found in the appendix.

**MADDPG:** The MADDPG algorithm is perhaps the most popular general-purpose off-policy MARL algorithm. The algorithm was proposed by Lowe et al. (2017), based on the DDPG algorithm (Lillicrap et al., 2015), and uses a centralized Q-function taking observations and actions from all the agents to alleviate the non-stationarity issue and stabilize multi-agent training. Note that although DDPG was originally designed for continuous actions, MADDPG adopts the gumbel-softmax (Jang et al., 2016) trick to handle discrete actions.

**MATD3:** The TD3 algorithm (Fujimoto et al., 2018b) is a popular enhanced version of the standard DDPG algorithm (Lillicrap et al., 2016). We similarly apply the same centralized critic technique introduced in MADDPG to TD3 to derive a multi-agent variant of TD3, i.e., MATD3. The only difference between MATD3 and MADDPG is the use of twin delayed critics and the addition of small amounts of noise to the actions sampled from the buffer.

**MASAC:** The Soft Actor-Critic (SAC) algorithm (Haarnoja et al., 2018) is an extremely popular off-policy algorithm and has been considered as a state-of-the-art baseline for a diverse range of RL problems with continuous actions. Similar to MADDPG, we introduce a centralized critic in SAC to achieve another general-purpose MARL algorithm, MASAC.

**QMix:** QMix (Rashid et al., 2018) is a Q-learning algorithm designed for multi-agent *cooperative* tasks with a global reward. The core idea of QMix is value decomposition, which formulates the global Q function, $Q_{tot}$ as the output of a "mixer" neural network whose inputs are the individual agent Q functions, $Q_a$; The weights of this "mixer" network are constrained to be positive in order to insure that $\frac{\partial Q_{tot}}{\partial Q_a} \geq 0, \forall$ agents $a$. This ensures that by acting greedily to maximize their local Q functions, agents will also be maximizing the global Q function. QMix was first introduced in the StarCraftII micromanagement and has been a popular benchmark algorithm for this challenge. However, it is rarely tested in other domains.

**MAPPO:** In addition to the off-policy algorithms above, we also consider an on-policy MARL algorithm, i.e., a multi-agent variant of PPO (MAPPO). We enhance the standard decentralized PPO algorithm by learning a centralized critic that takes in the global state or the concatenation of each agent's local observation (when global state is not available). This idea of centralized value functions was originally introduced in the COMA algorithm (Foerster et al., 2018). Our implementation follows the details in Baker et al. (2019), including using two separate networks for policy and value function, GAE, advantage normalization and a Pop-Art value normalizer (Hessel et al., 2019), which uses running statistics of the values to normalizes the regression target of value network to zero mean and a standard deviation of one.

### 3.3 KEY IMPLEMENTATION DETAILS

We highlight some important techniques here. More details can be found in Appendix Sec. B.

To increase the likelihood that differences in performance are primarily coming from the algorithm, we ensure all the policies use the same network architecture for both actor and critic. We use parameter sharing for all agents, i.e., all the agents share the same weights.

To handle partial observability, recurrent networks are used in all the algorithms. For MAPPO, we follow Baker et al. (2019), which cuts a full trajectory into small chunks (i.e., 10 timesteps per chunk) and maintains the initial hidden state of each chunk. Then the RNN policy is trained over a batch of data chunks. A similar chunking technique has also been introduced for off-policy learning by Kapturowski et al. (2018), which, however, does not work well in our experiments. Instead, we follow the procedure in Rashid et al. (2020) and Hausknecht & Stone (2015), which trains the recurrent policy over a batch of complete trajectories. More specifically, for each timestep $t$ in a training trajectory, we start with a zero-vector as the initial hidden state and re-feed all timesteps up to $t$ to make sure the hidden state of time $t$ is up-to-date.

Lastly, different from standard multi-agent games where all the agents act at the same time, the Hanabi challenge is a *turn-based* game, i.e., all the agents take actions one by one. Hence, in an $N$-player Hanabi game, we decompose a trajectory into $N$ separate sub-trajectories for each agent and each agent's policy is only trained on its own state-action sequence. We make sure that for each agent's training trajectories, all the other agents' actions are properly observed in the state.

## 4 ENVIRONMENTS

We briefly outline the settings used to test the algorithms as well as key details on the modifications we made to their original implementations. We will refer to each of these settings as *domains* and each of the individual problems in the domains as *environments*. From each *domain* we only include the fully cooperative tasks.

**Multi-agent Particle World** environments were introduced in Lowe et al. (2017). These environments consist of various multi-agent games in a 2D world with small particles navigating within a square box. We will refer to these environments as the *MPEs*, i.e., multi-particle environments. We consider the 3 fully cooperative tasks from the original set shown in Fig. 1(a): *simple-spread*, *speaker-listener*, and *simple-reference*. Note that since the two agents in *speaker-listener* have different observation and action spaces, this is the only setting in this paper where we do not share parameters but train separate policies for each agent.

As will be discussed in Sec. 5, almost all algorithms solve all of the MPE tasks, so we primarily use it as a basic check that our algorithms are likely correctly implemented.

**StarCraftII Micromanagement** (SMAC) [1] was introduced in Samvelyan et al. (2019). In these tasks, decentralized agents must cooperate to defeat adversarial bots in various scenarios with a wide range of agent numbers (from 2 to 27). We use the global game state to train our centralized critics or Q-functions. Note that an agent may "die" in an episode. We masked out the inputs of dead agents by zero and make sure they do not contribute to the learning objective. Fig. 2(a) (left) shows two example StarCraftII environments.

**Hanabi** is a turn-based card game, introduced as a MARL challenge in Bard et al. (2020), where each agent observes other players' cards except their own cards. A visualization of the game is shown in Fig. 1(b). The goal of the game is to send information tokens to others and cooperatively take actions

---

[1] Version 4.10 is used in this paper.

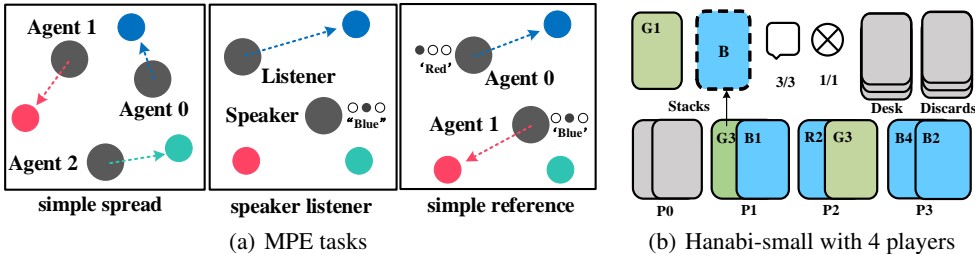

(a) MPE tasks   (b) Hanabi-small with 4 players

Figure 1: Task visualizations. (a) the *Particle-World* domain. *simple-spread* (left): agents need to cover all the landmarks and do not have a color preference for the landmark they navigate to; *speaker-listener* (middle): the listener needs to navigate to a specific landmarks following the instruction from the speaker; *simple-reference* (right): both agents only know the other's goal landmark and needs to communicate to ensure both agents move to the desired target. (b) The simplified Hanabi domain (*Hanabi-small*): We use just two colors instead of the five used in the original Hanabi game.

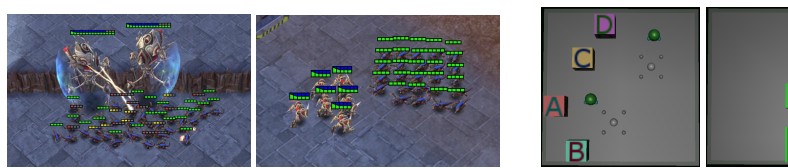

(a) Two representative StarCraftII micromanagement scenarios (*2c_vs_64zg* and *corridor*).

(b) Tasks in the hide-and-seek domain: *construction*, *box-locking* and *box-locking-easy*.

Figure 2: Visualizations of the StarCraftII domain and the hide-and-seek domain.

to stack as many cards as possible in ascending order to collect points. Since the original full Hanabi game may require up over 100 million samples for a general-purpose algorithm to converge (Foerster et al., 2019), we consider a simplified variant, "*Hanabi-small*" from Bard et al. (2020), which only has 2 colors, hand size 2, 1 life token, 3 information tokens and a maximum score of 10.

**The Hide-And-Seek Domain**, introduced by Baker et al. (2019), is a MuJoCo-based physical world with agents, walls and manipulable objects. We modify the original environment to generate three cooperative games adapted from the transfer task set in Baker et al. (2019), including *box-locking*, where agents need to lock all the boxes as fast as possible, and *construction*, where agents need to manipulate boxes to desired locations. We also consider a simplified version of *box-locking*, named *box-locking-easy*, with a much smaller maze size for faster training and hyperparameter tuning. Note that in *box-locking*, all the boxes need to be locked for a success while in *construction*, all the boxes have to be placed on the target sites for a success. See visualizations of the environments in Fig. 2(b).

## 5 EXPERIMENT RESULTS

### 5.1 EXPERIMENTAL DETAILS

To ensure a fair comparison over different algorithms, we (1) make sure that all policy architectures and weight initialization are consistent across algorithms (2) use similarly sized grid searches for all algorithms with slightly larger grid searches for the worst performing algorithms. For the algorithms we tune the following hyperparameters and report the values and sweeping procedure in Appendix Sec. B.1:

- MADDPG, MATD3: learning rate and the polyak update rate for target-network updates.
- MASAC: learning rate, polyak update rate for target networks, target entropy coefficient.
- QMix: learning rate, target network update frequency.
- MAPPO: learning rate, epoch.

These algorithms, particularly the on-policy MAPPO algorithm, are highly sensitive to batch size. In domains other than the MPEs, where batch size has a nominal effect, we use the largest possible batch-size possible that can be fit into GPU memory. Every single environment is conducted in a single desktop machine using a single GPU card for training. To enable easy comparison of wall-clock time between algorithms, we run all of the algorithms without any parallelism in the environment. For each experiment, we keep training until convergence or at most 3 days. All the results are averaged

| scenarios | MAPPO | QMix | MASAC | MATD3 | MADDPG |
|---|---|---|---|---|---|
| simple-spread | -122.131 | -127.95 | -129.497 | -142.907 | -156.161 |
| speaker-listener | -12.3095 | -13.3582 | -13.098 | -34.9912 | -31.0721 |
| simple-reference | -9.84231 | -13.9181 | -13.1636 | -44.4698 | -49.3984 |

Table 1: Average episode rewards in MPE.

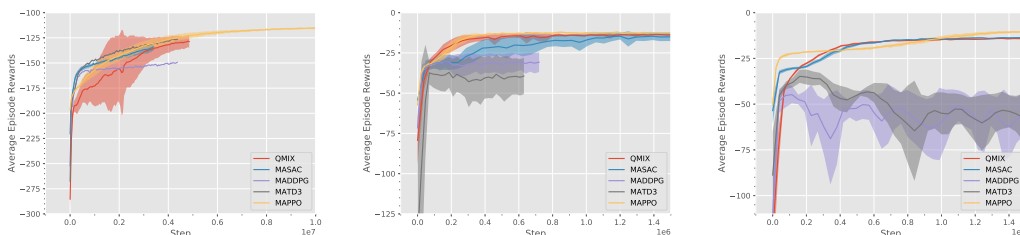

(a) Learning curves of different algorithms w.r.t. environment steps.

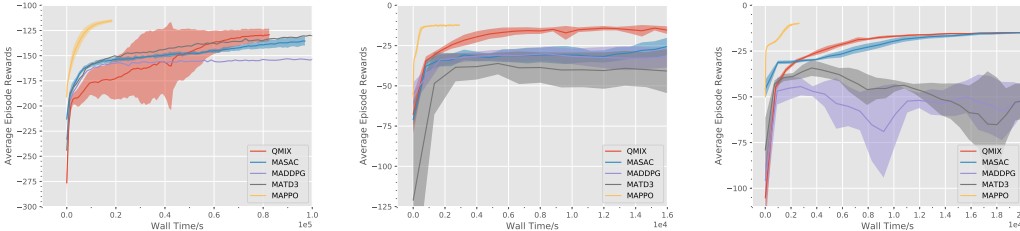

(b) Learning curves of different algorithms w.r.t. running time.

Figure 3: Evolution of cumulative episode rewards according to step-number (top) and wall-clock time (bottom) for different algorithms in the particle-world domain.

over 5 random seeds in the StarCraftII domain, Hanabi and Hide-And-Seek domain, and over 10 seeds in the particle-world domain.

For hyperparameter optimization we perform a sequential search per hyperparameter, optimizing them one at a time and then proceeding onto the next hyper-parameter using the best value found previously. For example, we might tune learning rate, find the best learning rate, and then move on to optimizing target update frequency while keeping learning rate fixed to the best value. We perform this procedure on one representative environment in each domain and re-use the hyperparameters for the other environments.

The only exception to our outlined hyperparameter tuning is QMix in the StarCraftII domain, where we simply inherit the hyperparameters from the original QMix paper as these parameters are already tuned for good performance. For a similar reason, in these environments we re-use the QMix batch size of 32 episodes for all the off-policy algorithms in the StarCraftII domain.

## 5.2 PARTICLE-WORLD

**Hyperparameter Selection:** The hyperparameters are tuned on the *simple-spread* scenario for each algorithm and reused across the other two scenarios.

**Asymptotic reward:** The final average scores at convergence are shown in Table. 1 We find that all of the algorithms perform somewhat similarly, with some degradation in performance for MATD3 and MADDPG in the speaker-listener and simple-reference scenarios. Note that the results for MADDPG we obtained are similar to the numbers reported in the original MADDPG paper.

**Wall-clock time:** Fig. 3 shows the evolution of the rewards both in step number and wall-clock time. The run-time is dominated by gradient steps rather than environment collection and consequently we observe that MAPPO, which takes many fewer gradient steps, converges to a good solution much quicker than the other algorithms. Note that we have turned off the parallel sample collection in MAPPO. Hence, with enough computational resources, the wall-clock time of MAPPO could be even lower.

| Maps | MAPPO | QMix | MASAC | MATD3 | MADDPG |
|---|---|---|---|---|---|
| 2m_vs_1z | 100 | 100 | 90.88 | 89.84 | 89.06 |
| 3m | 99.875 | 94.71 | 55.54 | 60.125 | 48.63 |
| 2s_vs_1sc | 99.56 | 95.25 | 20.06 | 0 | 46.63 |
| 2s3z | 98.5 | 94.07 | 23.52 | 3.125 | 14.59 |
| 3s_vs_3z | 99.5625 | 97.03 | 0 | 0 | 0 |
| 3s_vs_4z | 99.27 | 95.98 | 0 | 0 | 0 |
| 2c_vs_64zg | 99.375 | 96.08 | 0 | 0 | 0 |
| so_many_baneling | 99.68 | 93.18 | 62.43 | 45.28 | 56.15 |
| 8m | 99.48 | 95.07 | 43.65 | 0 | 0 |
| MMM | 98.59 | 96.95 | 16.83 | 0 | 0 |
| 1c3s5z | 100 | 96.09 | 24 | 0 | 0 |
| 3s5z_vs_3s6z | 80 | 79.03 | 0 | 0 | 0 |
| bane_vs_bane | 94.4792 | 91.97 | 0 | 0 | 0 |
| 25m | 96.88 | 86.15 | 0 | 0 | 0 |
| 5m_vs_6m | 75.01 | 71.46 | 0 | 0 | 0 |
| 8m_vs_9m | 92.97 | 93.44 | 0 | 0 | 0 |
| 10m_vs_11m | 84.27 | 94.85 | 0 | 0 | 0 |
| 3s5z | 90.94 | 82.15 | 0 | 0 | 0 |
| MMM2 | 0 | 86.43 | 0 | 0 | 0 |
| 3s_vs_5z | 23.35 | 96.58 | 0 | 0 | X |
| 6h_vs_8z | 8.59 | 79.32 | 0 | 0 | 0 |
| corridor | 0 | 75.625 | 0 | 0 | 0 |
| 27m_vs_30m | 10.625 | 64.06 | 0 | 0 | 0 |

Table 2: Eval Win rate in SMAC maps over 32 trials.

## 5.3 StarCraftII Micomanagement

**Hyperparameter Selection:** We use the easier maps **3m** and **2s3z** to do the hyperparameter tuning where easier is characterized both by how quickly QMix finds a solution as well as how many agents are involved. For MAPPO we found that all hyperparameters worked for **3m** so we used **2s3z** for the tuning whereas for MASAC/MATD3/MAaDDPG we could not maximize performance on **3m**, the easiest map, so we stuck to tuning parameters on that map. As mentioned before, for QMix we simply re-use the hyperparameters from the original QMix paper (Rashid et al., 2020). We run each of the maps for 10 million steps. Note that this is longer than in the QMix paper (Rashid et al., 2020) and so may lead to better results than are reported there.

We evaluate the algorithms on 32 testing episodes and consider the test winning rate as our performance metric. Table. 2 represents the performance metric for all of our algorithms. The results demonstrate that in easy and medium environments, MAPPO performs as well, if not better, than QMix. However, in the hardest maps, MAPPO underperforms in relation to QMix. A deeper investigation of this trend revealed that with MAPPO's performance in these difficult maps significantly improves with a lower number of epochs - specifically, 5 epochs instead of 15. Appendix A.1 shows the results for MAPPO with 5 epochs. This may be caused by the fact that a higher epoch number results in a larger amount of data reuses, potentially rendering the hidden states of the recurrent network stale. We also note that the off-policy actor critic algorithms, while showing promise in the easiest maps, quickly drop in performance on the medium and difficult maps, in which they consistently fail.

## 5.4 Hanabi

For Hanabi, we tune hyperparameters on the 2-player game and re-use the chosen hyperparameters on the remaining games. We trained all algorithms for either 100 million steps or 3 days, whichever came first.

We report the averaged training rewards of the best hyperparameters in Table. 3. No hyperparameters for the off-policy algorithms recieved a score above one despite the same hyperparameter grid working for other domains. Since the task only gets harder as we increase agent number, we only report the results for increasing agent number for MAPPO.

|  | MAPPO | QMix | MASAC | MATD3 | MADDPG |
|---|---|---|---|---|---|
| 2 players | 6.8777 | 0.2907 | 1.094 | 0.1026 | 0.4211 |
| 3 players | 5.1788 | X | X | X | X |
| 4 players | 3.9557 | X | X | X | X |
| 5 players | 3.5 | X | X | X | X |

Table 3: Average score in the *Hanabi-Small* with different number of players. X's represent trials that were not run due to underperformance.

## 5.5 The Hide-And-Seek Domain

We tune hyperparameters on the *box-locking-easy* environment and the chosen hyperparameters are re-used on the other environments. We report the final success rate for each of the tasks in Table. 4. We find that MAPPO performs well across the environments, while of the off-policy algorithms, only MASAC has high performance on the easy task. Even after 2 days of training, MASAC was not able to solve the the two harder tasks. However, MAPPO is able to achieve its results within half a day of training time despite operating without parallelism.

| Tasks | MAPPO | QMix | MASAC | MATD3 | MADDPG |
|---|---|---|---|---|---|
| box-locking-easy | 97.0 | 10.6 | 93.0 | 56.5 | 73.23 |
| box-locking | 96 | 0 | 2.3 | 0.0 | 0 |
| construction | 48.0 | 0 | 9 | 0 | 0.2 |

Table 4: Success rate of each of the algorithms in the Hide-And-Seek domain.

## 5.6 Summary

Across environments we find that MAPPO has strong average performance on all domains while being occasionally outperformed by other algorithms on a few environments. More problematically, we find that we cannot easily tune MADDPG/MATD3/MASAC outside of the particle environments and that the same issue applies to QMix outside of the SMAC environments. These issues could probably be alleviated with a more extensive grid search for the off-policy algorithms but suggest either that 1) MAPPO is a more stable algorithm across environments or 2) that the default PPO hyperparameters around which we search are well configured for MARL tasks.

## 6 Conclusion

The results in Sec. 5 suggest that MAPPO is a good starting point for tackling new MARL problems. Although QMix occasionally outperforms PPO on certain maps in the StarCraftII domain, this is achieved using tuned hyperparameters from the original implementation and MAPPO might achieve similar performance given equal tuning budget. Given access to equally sized compute budgets for hyperparameter optimization, MAPPO is the most consistently successful algorithm across the studied environments, performing well at SMAC, as well as any algorithm in the MPEs and the best in hide-and-seek and Hanabi-small. Surprisingly, we also find that the centralized Q function variants DDPG, SAC, and TD3 almost entirely fail to solve any of the studied tasks, particularly in slightly harder environments.

Of course, these results are likely dependent on the choice of hyperparameters and architectural choices. Thus, it may be the case that the initial point around which we perform our grid search for MADDPG/MATD3/MASAC does not contain any good solutions and that performance of a set of hyperparameters on the MPEs is not predictive of performance in other domains. Finding a better set of hyperparameters over which to search for the off policy algorithms is an important direction for future work. Similarly, seeing if our conclusions hold over a larger set of problem domains would additionally be valuable. It is possible that our conclusions are a function of the restricted length of time we ran the algorithms for; it would be interesting to investigate the asymptotic performance of these algorithms over much longer training times. Finally, it would be worthwhile to investigate the failures of these off-policy algorithms in these settings to understand more deeply the performance discrepancy.

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

| common hyperparameters | value |
|---|---|
| ppo epoch | 15 |
| recurrent data chunk length | 10 |
| use clipped_value_loss | True |
| max clipped value loss | 0.2 |
| entropy coef | 0.01 |
| gradient clip norm | 10.0 |
| gae lamda | 0.95 |
| gamma | 0.99 |
| loss | huber loss |
| huber delta | 10.0 |
| batchsize | parallel envs $\times$ episode length $\times$ num_agents |
| mini_batch_size | batch_size / num_mini_batch |

Table 5: common hyperparameters used in MAPPO across all domains.

# A    TRAINING DETAILS

## A.1    ALGORITHM HYPER-PARAMETERS

Below, we list the details of the hyperparameters used for all experiments, as well as the hyperparameter sweeping procecure for hyperparameters that were tuned.

For MAPPO, certain hyperparameters were kept constant across all environments; these are listed in Tab. 5, and are mostly inherited from the baseline implementation of PPO. Note that since we use parameter sharing and mix all agents' data, the actual batch-size will be larger with more agents. Regarding the mini-batch-size, we use the largest possible mini-batch-size under the constraint of GPU memory limit. Note that while we list "parallel envs" for MAPPO, we actually run it without any parallelism to create fair comparisons of wallclock speed between algorithms. For MAPPO, "parallel envs" is only used to control the batch size.

**Network architecture hyperparameters:** All neural networks are implemented with the following overall structure: $m$ fully connected layers, followed by $k$ Gated Recurrent Unit (GRU) (Chung et al., 2014) layers, and lastly by $n$ fully connected layers. In the following hyperparameter tables, the "num fc before" hyperparameter corresponds to $m$, "num GRU layers" corresponds to $k$, and "num fc after" corresponds to $n$. The nonlinearity function used is the rectified linear unit (ReLU) (Nair & Hinton, 2010), with a gain value equal to the "ReLU gain" hyperparameter. Lastly, The "use feature normalization" hyperparameter describes if layer normalization (Ba et al., 2016) is applied to the inputs.

**Tuned hyperparameters:** For MADDPG, MATD3, and MASAC, the "tau" hyperparameter corresponds to the rate of the polyak average technique used to update the target networks. For MASAC, the "target entropy coef" hyperparameter determines the target entropy; specifically, the target entropy is given by: $\bar{H} = \beta * (-log(\frac{1}{|A|}))$, where $|A|$ is the dimension of the discrete action space, and $\beta$ is equal to the "target entropy coef" hyperparameter. For QMix, the "hard target update interval" hyperparameter specifies the number of gradient updates which must elapse before the target network parameters are updated to equal the live network parameters. *Note that the tables specifying the sweeping procedure for each domain list the hyperparameters in the order in which they were tuned.*

For each domain, we describe the network architectures, the hyperparameters kept constant across that domain for each algorithm, as well as the hyperparameter sweeping procedure for tunable hyperparameters for each algorithm.

**Multi-agent Particle World (MPE):**

| | |
|---|---|
| QMix | lr [1e-4, 5e-4, **7e-4**, 1e-3]; hard update interval [**200**, 500, 700, 900, 1500] |
| MADDPG | lr [1e-4, 5e-4, **7e-4**, 1e-3, 1e-2]; tau [0.01, **0.005**, 0.001] |
| MATD3 | lr [1e-4, **5e-4**, 7e-4, 1e-3, 1e-2]; tau [0.01, **0.005**, 0.001] |
| MASAC | lr [1e-4, 5e-4, **7e-4**, 1e-3, 1e-2]; tau [0.01, **0.005**, 0.001]; target entropy coef [0.1, **0.3**, 0.5, 0.7] |
| MAPPO | lr [1e-3, **7e-4**, 5e-4, 1e-4]; ppo epoch [5, 10, **15**, 20, 25] |

Table 6: sweeping procedure in MPE, tau denotes the target network update rate, the bold font indicates the value adopted.

| network hyperparameters | value |
|---|---|
| initialization method | orthogonal |
| num GRU layers | 1 |
| RNN hidden state dim | 64 |
| fc layer dim | 64 |
| num fc before | 2 |
| num fc after | 1 |
| use feature normalization | True |
| optimizer | Adam |
| optimizer eps | 1e-5 |
| weight decay | 0 |
| ReLU | True |
| ReLU gain | sqrt(2) |

Table 7: network hyperparameters used in the MPE domain by all algorithms

| hyperparameters | value |
|---|---|
| episode length | 25 |
| last action layer gain | 0.01 |
| gamma | 0.99 |
| buffer size | 5000 |
| batch size | 32 |
| epsilon | from 1.0 to 0.05 |
| epsilon anneal time | 50000 |
| Q function loss | MSE loss |

Table 8: Hyperparameters used in the MPE domain by MADDPG, MATD3, MASAC, and QMix. Buffer size and batch size is calculated by episodes.

| hyperparameters | value |
|---|---|
| parallel envs | 128 |
| last action layer gain | 0.01 |
| num mini batch | 1 |

Table 9: Hyperparameters used in the MPE domain by MAPPO

**StarCraftII Micromanagement (SMAC)**: All algorithms are trained until convergence, 10M timesteps is reached, or for a maximum of 3 days. Since the maximum length of an episode varies per map, we change the maximum episode length based on the maximum amount allowed by the map.

| | |
|---|---|
| QMix | use open-source hyperparameters |
| MADDPG | lr [1e-3, **5e-4**, 1e-4] tau [0.01, **0.005**, 0.001] |
| MATD3 | lr [1e-3, **5e-4**, 1e-4] tau [0.01, **0.005**, 0.001] |
| MASAC | lr [1e-3, **5e-4**, 1e-4] tau [0.01, **0.005**, 0.001] target entropy coef [0.1, **0.3**, 0.5, 0.7] |
| MAPPO | lr [1e-3, 7e-4, **5e-4**, 1e-4], ppo epoch [**5**, 10, **15**, 20, 25] |

Table 10: Sweeping procedure in the SMAC domain, the bold font indicates the value adopted.

| network hyperparameters | value |
|---|---|
| initialization method | orthogonal |
| num GRU layers | 1 |
| RNN hidden state dim | 64 |
| fc layer dim | 64 |
| num fc before | 2 |
| num fc after | 1 |
| use feature normalization | True |
| optimizer | Adam |
| optimizer eps | 1e-5 |
| weight decay | 0 |
| ReLU | True |
| ReLU gain | sqrt(2) |

Table 11: network hyperparameters used in the StarCraft domain by all algorithms

| hyperparameters | value |
|---|---|
| episode length | depends on maps |
| last action layer gain | 1 |
| gamma | 0.99 |
| buffer size | 5000 |
| batch size | 32 |
| epsilon | from 1.0 to 0.05 |
| epsilon anneal time | 50000 |
| Q-function loss | MSE loss |

Table 12: Hyperparameters used in SMAC by MADDPG, MATD3, MASAC, and QMix. buffer size and batch size is calculated by episodes.

| hyperparameters | value |
|---|---|
| parallel envs | 8 |
| episode length | 400 |
| last action layer gain | 1 |

Table 13: Hyperparameters used in the SMAC domain by MAPPO

| Maps | MAPPO | QMix | MASAC | MATD3 | MADDPG |
|---|---|---|---|---|---|
| | | epoch=15 | | | |
| 2m_vs_1z | 100 | 100 | 90.88 | 89.84 | 89.06 |
| 3m | 99.875 | 94.71 | 55.54 | 60.125 | 48.63 |
| 2s_vs_1sc | 99.56 | 95.25 | 20.06 | 0 | 46.63 |
| 2s3z | 98.5 | 94.07 | 23.52 | 3.125 | 14.59 |
| 3s_vs_3z | 99.5625 | 97.03 | 0 | 0 | 0 |
| 3s_vs_4z | 99.27 | 95.98 | 0 | 0 | 0 |
| 2c_vs_64zg | 99.375 | 96.08 | 0 | 0 | 0 |
| so_many_baneling | 99.68 | 93.18 | 62.43 | 45.28 | 56.15 |
| 8m | 99.48 | 95.07 | 43.65 | 0 | 0 |
| MMM | 98.59 | 96.95 | 16.83 | 0 | 0 |
| 1c3s5z | 100 | 96.09 | 24 | 0 | 0 |
| 3s5z_vs_3s6z | 80 | 79.03 | 0 | 0 | 0 |
| 8m_vs_9m | 92.97 | 93.44 | 0 | 0 | 0 |
| bane_vs_bane | 94.4792 | 91.97 | 0 | 0 | 0 |
| 25m | 96.88 | 86.15 | 0 | 0 | 0 |
| 5m_vs_6m | 64.58 | 71.46 | 0 | 0 | 0 |
| 6h_vs_8z | 8.59 | 79.32 | 0 | 0 | 0 |
| corridor | 0 | 75.625 | 0 | 0 | 0 |
| | | epoch=5 | | | |
| 3s_vs_5z | 96.88 | 96.58 | 0 | 0 | 0 |
| 3s5z | 96.84 | 82.15 | 0 | 0 | 0 |
| MMM2 | 87.5 | 86.43 | 0 | 0 | 0 |
| 10m_vs_11m | 87.5 | 94.85 | 0 | 0 | 0 |
| | | lr=7e-4,epoch=5 | | | |
| 27m_vs_30m | 61.25 | 64.06 | 0 | 0 | 0 |

Table 14: Eval win rate in SMAC domain over 32 trials.

**Hanabi**

| QMix | lr[1e-3, 5e-4, 1e-4, **2.5e-5**]; hard update interval [200, 500, **900**, 1500] |
|---|---|
| MADDPG | lr[1e-3, **5e-4**, 1e-4, 2.5e-5]; tau [0.001, **0.005**, 0.01] |
| MATD3 | lr[1e-3, **5e-4**, 1e-4, 2.5e-5]; tau [0.001, **0.005**, 0.01] |
| MASAC | lr[1e-3, **5e-4**, 1e-4, 2.5e-5]; tau [0.001, **0.005**, 0.01]; target entropy coef [**0.1**, 0.3, 0.5, 0.7] |
| MAPPO | lr[1e-3, **7e-4**, 5e-4, 1e-4]; ppo epoch [5, 10, **15**, 20, 25] |

Table 15: Sweeping procedure in the Hanabi domain, the bold font indicates the value adopted.

| network hyperparameters | value |
|---|---|
| initialization method | orthogonal |
| num GRU layers | 1 |
| RNN hidden state dim | 512 |
| fc layer dim | 64 |
| num fc before | 3 |
| num fc after | 1 |
| use feature normalization | False |
| optimizer | Adam |
| optimizer eps | 1e-5 |
| weight decay | 0 |
| ReLU | True |
| ReLU gain | sqrt(2) |

Table 16: Network hyperparameters used in the Hanabi domain by all algorithms

| hyperparameters | value |
|---|---|
| episode length | 80 |
| last action layer gain | 0.01 |
| gamma | 0.99 |
| buffer size | 5000 |
| batch size | 64 |
| epsilon | from 1.0 to 0.05 |
| epsilon anneal time | 80000 |
| Q function loss | MSE loss |

Table 17: Hyperparameters used in the Hanabi domain by MADDPG, MATD3, MASAC, and QMix. Buffer size and batch size is calculated by episodes.

| hyperparameters | value |
|---|---|
| parallel envs | 1000 |
| episode length | 80 |
| last action layer gain | 0.01 |

Table 18: Hyperparameters used in the Hanabi domain by MAPPO

**The Hide-And-Seek (HNS) domain**

| | |
|---|---|
| QMix | lr[1e-3, **5e-4**, 1e-4]; hard update interval [200, 500, 700, **900**] |
| MADDPG | lr[1e-3, **5e-4**, 1e-4]; tau [0.01, **0.005** 0.001] |
| MATD3 | lr[1e-3, **5e-4**, 1e-4]; tau [0.01, **0.005** 0.001] |
| MASAC | lr[1e-3, **5e-4**, 1e-4]; tau [0.01, **0.005**, 0.001] |
| MAPPO | lr[1e-3, **7e-4**, 5e-4]; ppo epoch [5, 10,**15**, 20, 25] |

Table 19: Sweeping procedure in the HNS domain

| network hyperparameters | value |
|---|---|
| initialization method | orthogonal |
| num GRU layers | 1 |
| RNN hidden state dim | 64 |
| fc layer dim | 64 |
| num fc before | 2 |
| num fc after | 1 |
| use feature normalization | True |
| optimizer | Adam |
| optimizer eps | 1e-5 |
| weight decay | 0 |
| ReLU | True |
| ReLU gain | sqrt(2) |

Table 20: Network hyperparameters used in the HNS domain by all algorithms

| hyperparameters | value |
|---|---|
| episode length | boxlocking: 120
blueprint construction: 200 |
| last action layer gain | 0.01 |
| gamma | 0.99 |
| buffer size | 5000 |
| batch size | 256 |
| epsilon | from 1.0 to 0.05 |
| epsilon anneal time | 50000 |
| Q function loss | MSE loss |

Table 21: Hyperparameters used in the HNS domain by MADDPG, MATD3, MASAC, and QMix. Buffer size and batch size is calculated by episodes.

| hyperparameters | value |
|---|---|
| parallel envs | boxlocking: 250
blueprint construction: 400 |
| episode length | 2 × task horizons
boxlocking: 240
construction:400 |
| last action layer gain | 0.01 |

Table 22: hyperparameters used in the HNS domain by MAPPO

## B   ALGORITHM DETAILS

We use *policies* to refer to each unique actor and critic network pair and *agents* to refer to the actors in the environment. We will use $m$ to denote the number of policies being trained, and $n$ to be the number of active agents in the environment. We will use $g(\circ)$ to refer to the function which maps each agent to the policy which controls it, and $A_i$ to be the number of agents under the control of policy $i$. In a setting where all agents share the same parameters, $m = 1$, as there is only a single policy, and $g$ would map all agents to this single policy. In a setting in which all agents have their own critic and actor, $m = n$, as the number of policies equals the number of agents, and $g$ would map each agent to its unique policy containing the actor and critic networks.

As a notational simplification, in all of the outlined algorithms we will assume that the environment and all agents have a fixed horizon length $T$; modifying the algorithms to work with varying time horizons is simple. We will also assume that the agents have discrete actions. For the MADDPG, MATD3, and MASAC algorithms, in order for actions to be differentiable in a discrete setting, we use the gumbel softmax trick Jang et al. (2016), which approximately samples an action from a categorical distribution parameterized by the output logits of the actor network. Finally, we will always assume that the number of active agents in the environment is fixed across the horizon; this lets us ensure that there is a constant length input to the centralized critic.

The psuedocode of MADDPG, MASAC, QMix and MAPPO with the support of recurrent policies are shown in Algo. 1, Algo. 2, Algo. 3 and Algo. 4 respectively.

**Algorithm 1:** MADDPG

Initialize $\theta_1, \cdots, \theta_m$, the parameters of agent critic networks, and $\phi_1, \cdots, \phi_m$, the parameters of the agent actor networks, using Orthogonal initialization (Hu et al., 2020);

Set the function $g$ mapping agent $a$ to index of the policy $i$ which controls the agent;

Set the learning rate $\alpha$, batch size $B$ and replay buffer $D = \{\}$;

$\theta_i^- = \theta_i, \phi_i^- = \phi_i$ for $i = 1...m$;

step=0, episodes=0;

**while** *step $\leq$ step$_{max}$* **do**

    $s_0 =$ initial state;

    initialize $h_0^{(1)} \cdots h_0^{(n)}$ actor RNN states;

    $\tau = \{\}$ empty trajectory;

    **for** *timestep t=1...T* **do**

        **for** *each agent a* **do**

            $i \leftarrow g(a)$ Get the index of the policy controlling agent $a$;

            $\epsilon = $ epsilon-schedule(step);

            $u_t^{(a)}, h_t^{(a)} = $

$$\begin{cases} \mu_i(o_t^{(a)}, h_{t-1}^{(a)}; \phi_i) \text{ using the Gumbel Softmax Trick} & \text{with probability } 1 - \epsilon \\ randint(1, |U|) & \text{with probability } \epsilon \end{cases}$$

        **end**

        Get reward $r_t$, next state $s_{t+1}$, done $d_t$, observations $o_t^1, \ldots, o_t^t$;

        $\tau = \tau \cup \{(s_t, \mathbf{u_t}, r_t, s_{t+1}, o_t^1, \ldots o_n^t)\}$;

        step $=$ step $+ 1$;

    **end**

    $D = D \cup \tau$;

    episodes $=$ episodes $+ 1$;

    **if** *episodes $\geq B$ **and** train-interval training steps have passed* **then**

        **for** *each policy i in 1...m* **do**

            b $\leftarrow$ random batch of B episodes from $D$ for policy $i$;

            **for** *each timestep t in each episode in batch b* **do**

                $Q_t = Q_i(s_t, u_t^{(1)}, ..., u_t^{(n)}, h_{Q,t-1}^{(i)}; \theta_i)$;

                Compute $\mu_{t+1}^{(a)} = \mu_{g(a)}(o_{t+1}^{(a)}, h_{\mu,t}^{'(a)}; \phi_{g(a)}^-)$ for each agent $a$;

                $Q_{t+1} = Q_i'(s_{t+1}, \mu_{t+1}^{(1)}, ..., \mu_{t+1}^{(n)}, h_{Q,t}^{'(i)}; \theta_i^-)$;

                $Q_t^* = r_t + (1 - d_t)\gamma(Q_{t+1})$;

                Update RNN states $h_{Q,t-1}^{(i)}, h_{Q,t}^{'(i)}, h_{\mu,t}^{'(j)}, j \in [1, n]$ using transitions from batch b

            **end**

            $\mathcal{L}(\theta_i) = \frac{1}{B}\frac{1}{T}\sum_k \sum_t (Q_t - Q_t^*)^2$;

            $\theta_i = \theta_i - \alpha\nabla_{\theta_i}\mathcal{L}(\theta_i)$;

            Reset RNN states $h_{Q,t}^{(i)}, h_{\mu,t}^{(i)}$;

            **for** *each timestep t in each episode in batch b* **do**

                **for** *each agent a controlled by policy i* **do**

                    $Q_t^{(a)} = Q(s_t, u_t^{(1)}, ..., \mu_i(o_t^{(a)}, h_{\mu,t-1}^{(a)}; \phi_i), ..., u_t^{(n)}, h_{Q,t-1}^{(i)}; \theta_i)$;

                    Update RNN states $h_{Q,t}^{(a)}, h_{\mu,t}^{(a)}$ using transitions from batch $b$

                **end**

            **end**

            $\mathcal{L}(\phi_i) = -\frac{1}{A_i}\frac{1}{B}\frac{1}{T}\sum_a \sum_k \sum_t Q_t^{(a)}$;

            $\phi_i = \phi_i - \alpha\nabla_{\phi_i}\mathcal{L}(\phi_i)$;

            $\theta_i^- = (1 - \lambda)\theta_i^- + \lambda\theta_i$;

            $\phi_i^- = (1 - \lambda)\phi_i^- + \lambda\phi_i$;

        **end**

    **end**

**end**

**Algorithm 2:** MASAC

Initialize $\theta_1, \cdots, \theta_m$, the parameters of agent critic networks, and $\phi_1, \cdots, \phi_m$, the parameters of the agent actor networks, using Orthogonal initialization (Hu et al., 2020);

Set the learning rate $\epsilon$, polyak update rate $\lambda$ and replay buffer $D = \{\}$;

Set the initial entropy temperature, $\alpha$, and the target entropy $\bar{H}$;

$\theta_i^- = \theta_i, \phi_i^- = \phi_i$ for $i = 1...m$;

step=0, episodes=0;

**while** *step* $\leq$ *step$_{max}$* **do**

    $s_0 =$ initial state, $\tau = \{\}$ empty trajectory;

    initialize $h_0^{(1)} \cdots h_0^{(n)}$ actor RNN states;

    **for** *timestep t=1...T* **do**

        **for** *each agent a* **do**

            $i \leftarrow g(a)$;

            $u_t^{(a)} \sim \pi_i(u_t | o_t^{(a)}, h_{t-1}^{(a)}; \phi_i)$;

            Update RNN state $h_{t-1}^{(a)}$ to $h_t^{(a)}$;

        **end**

        Get reward $r_t$, next state $s_{t+1}$;

        $\tau = \tau \cup \{(s_t, \mathbf{u_t}, r_t, s_{t+1})\}$;

        step = step + 1;

    **end**

    $D = D \cup \tau$;

    episodes = episodes + 1;

    **if** *episodes* $\geq B$ ***and*** *train-interval training steps have passed* **then**

        **for** *each policy i in 1...m* **do**

            b $\leftarrow$ random batch of $B$ episodes from $D$ for policy $i$;

            **for** *each timestep t in each episode in batch b* **do**

                $Q_t = Q(s_t, u_t^{(1)}, ..., u_t^{(n)}, h_{Q,t-1}^{(i)}; \theta_i)$;

                Compute $\bar{a}_{t+1}^{(a)} \sim \pi_{g(a)}(\bar{a}_{t+1} | o_{t+1}^{(a)}, h_{\pi,t}^{'(a)}; \phi_{g(a)}^-)$ for each agent $a$;

                $Q_{t+1} = Q_i(s_{t+1}, \bar{a}_{t+1}^{(1)}, ..., \bar{a}_{t+1}^{(n)}, h_{Q,t}^{'(i)}; \theta_i^-)$;

                $V_{t+1} = Q_{t+1} - \alpha \log(\pi_{g(a)}(\bar{a}_{t+1} | o_{t+1}^{(a)}, h_{\pi,t}^{'(a)}; \phi_{g(a)}^-))$;

                $Q_t^* = r_t + (1 - d_t)\gamma V_{t+1}$;

                Update RNN states $h_{Q,t-1}^{(i)}, h_{Q,t}^{'(i)}, h_{\pi,t}^{'(j)}, j \in [1, n]$ using transitions from batch b

            **end**

            $\mathcal{L}(\theta_i) = \frac{1}{B}\frac{1}{T}\sum_i \sum_t (Q_t - Q_t^*)^2$;

            $\theta_i = \theta_i - \epsilon \nabla_{\theta_i} \mathcal{L}(\theta_i)$;

            Reset RNN states $h_{Q,t}^{(i)}, h_{\pi,t}^{(i)}$;

            **for** *each timestep t in each episode in batch b* **do**

                **for** *each agent a controlled by policy i* **do**

                    $\bar{u}_t^{(a)} \sim \pi(\bar{u}_t | o_t^{(a)}, h_{\pi,t-1}^{(a)}; \phi_i)$ // Sample $\bar{u}_t^{(a)}$ using Gumbel Softmax Trick;

                    $Q_t^{(a)} = Q(s_t, u_t^{(1)}, ..., \bar{u}_t^{(a)}, ..., u_t^{(n)}, h_{Q,t-1}^{(i)}; \theta_i)$;

                    Update RNN states $h_{Q,t}^{(a)}, h_{\mu,t}^{(a)}$ using transitions from batch b

                **end**

            **end**

            $\mathcal{L}(\phi_i) = \frac{1}{A_i}\frac{1}{B}\frac{1}{T}\sum_a \sum_k \sum_t \alpha \log(\pi(\bar{u}_t | o_t^{(a)}, h_{\pi,t-1}^{(a)}; \phi_i)) - Q_t$;

            $\phi_i = \phi_i - \epsilon \nabla_{\phi_i} \mathcal{L}(\phi_i)$;

            $\theta_i^- = (1 - \lambda)\theta_i^- + \lambda\theta_i$;

            $\phi_i^- = (1 - \lambda)\phi_i^- + \lambda\phi_i$ ;

            $\mathcal{L}(\alpha) = \frac{1}{A_i}\frac{1}{B}\frac{1}{T}\sum_k \sum_t -\alpha[\log(\pi(\bar{u}_t | o_t^{(a)}, h_{\pi,t-1}^{(a)}; \phi_i)) + \bar{H}]$;

            $\alpha = \alpha - \epsilon \nabla_\alpha \mathcal{L}(\alpha)$

        **end**

    **end**

**end**

---

**Algorithm 3:** QMix

---

Initialize $\theta_h$, the hypernetwork parameters, and $\phi_i$, the parameters of the agent Q networks for
  policy $i$, for $i \in \{1 \cdots m\}$, using Orthogonal initialization (Hu et al., 2020);
Set $\theta = \{\theta_h, \phi_1, ...\phi_m\}$, the collection of all parameters.;
Set $\theta^- = \theta$;
Set the learning rate $\alpha$, and replay buffer $D = \{\}$;
$step = 0, episodes = 0$;
**while** *step $\leq step_{max}$* **do**
    $s_0$ = initial state, $\tau = \{\}$ empty trajectory;
    initialize $h_0^{(1)} \cdots h_0^{(n)}$ actor RNN states;
    **for** *timestep t=1...T* **do**
        **for** *each agent a* **do**
            $i \leftarrow g(a)$;
            $\epsilon$ = epsilon-schedule(step);
$$u_t^{(a)}, h_t^{(a)} = \begin{cases} \arg\max_{u_t^{(a)}} Q_i(o_t^{(a)}, u_t^{(a)}, h_{t-1}^{(a)}; \phi_i) & \text{with probability } 1 - \epsilon \\ \text{randint}(1, |U|) & \text{with probability } \epsilon \end{cases}$$
        **end**
        Get reward $r_t$, next state $s_{t+1}$;
        $\tau = \tau \cup \{(s_t, \mathbf{u_t}, r_t, s_{t+1})\}$;
        step = step + 1;
    **end**
    $D = D \cup \tau$;
    episodes = episodes + 1;
    **if** *episodes $\geq$ B **and** train-interval training steps have passed* **then**
        $b \leftarrow$ random batch of B episodes from D;
        **for** *each timestep t in each episode in batch b* **do**
            $Q_t = $
              Mixer-net($Q_{g(1)}(o_t^{(1)}, h_{t-1}^{(1)}; \phi_{g(1)}), ..., Q_{g(n)}(o_t^{(n)}, h_{t-1}^{(n)}; \phi_{g(n)});$ hypernet($s_t; \theta_h$));
            $Q_{t+1} = $
              Mixer-net($Q_{g(1)}(o_{t+1}^{(n)}, h_t^{'(1)}; \phi_{g(1)}^-), ..., Q_{g(n)}(o_{t+1}^{(n)}, h_t^{'(n)}; \phi_{g(n)}^-);$ hypernet($s_{t+1}; \theta_h^-$));

            $Q_t^* = r_t + (1 - d_t)\gamma Q_{t+1}$;
            Update RNN states $h^{(i)}, h'^{(i)}, i \in [1, n]$ using transitions from batch b
        **end**
        $\mathcal{L}(\theta) = \frac{1}{B}\frac{1}{T}\sum_i\sum_t(Q_t^* - Q_t)^2$;
        $\theta = \theta - \alpha\nabla_\theta\mathcal{L}(\theta)$
    **end**
    **if** *update-interval training steps have passed* **then**
        $\theta^- = \theta$
    **end**
**end**

---

---

**Algorithm 4:** MAPPO

---

Initialize $\theta_i$, the parameter for policy $i$ and $\phi_i$, the parameter for critic $i$, for $i \in \{1 \cdots m\}$, using
  Orthogonal initialization (Hu et al., 2020);
Set the learning rate $\alpha$;
**while** *step $\leq$ step$_{max}$* **do**
    set data buffer $D = \{\}$;
    **for** *batchsize b=1...B* **do**
        $\tau = []$ empty trajectory list;
        initialize $h_0^{(1)} \cdots h_0^{(n)}$ actor RNN states;
        **for** *timestep t=1...T* **do**
            **for** *each agent a* **do**
                $i \leftarrow g(a)$;
                $u_t^{(a)}, h_t^{(a)} = \pi^{(a)}(s_t, h_{t-1}^{(a)}; \theta_i)$;
                $v_t^{(a)} = V^{(a)}(s_t, h_{t-1}^{(a)}; \phi_i)$ ;
            **end**
            Get reward $r_t$, next state $s_{t+1}$;
            $\tau = \tau + \{[(s_t, \mathbf{u_t}, \mathbf{h_t}, \mathbf{v_t}, r_t, s_{t+1})]\}$;
        **end**
        compute advantage $A$ and value target $V'$ for $\tau$ by applying GAE on $\tau$;
        // process a trajectory into chunks of length 10;
        **for** *timestep $t = 1, 11, 21, \ldots, T - 9$* **do**
            $D = D \cup (\tau[t : t + 10], A[t : t + 10], V'[t : t + 10])$;
        **end**
    **end**
    normalize all the advantages in $D$ to zero mean and one standard deviation;
    **for** *mini-batch k=1...K* **do**
        **for** *each policy i=1...n* **do**
            $b \leftarrow$ random mini-batch from $D$ for policy $i$;
            **for** *each data chunk c in the mini-batch b* **do**
                update the RNN hidden states for each timestep in $c$ from the first hidden state;
            **end**
            Adam update for $\theta_i$ with PPO objective and data $b$;
            Adam update for $\phi_i$ with value regression objective and data $b$;
        **end**
    **end**
**end**

---

