# OpenReview forum: "Benchmarking Multi-Agent Deep Reinforcement Learning Algorithms"
_ICLR.cc/2021/Conference — Reject_

### Official Review · AnonReviewer4 · 2020-10-27
**Useful benchmark but limited significance (so far)**

**Rating:** 4
**Confidence:** 3

**Review:**

## Summary
The paper provides a useful benchmark (a suite of cooperative multi-agent RL tasks) and a nice comparison of common as well as uncommon-but-successful algorithms on these tasks. A limited budget of hyperparameter tuning is applied to each algorithm in order to give each one a chance at good performance, and a clear winner emerges for consistent strong performance across tasks.

## Quality & Clarity
The paper is presented according to a high standard of quality. They document and explain their methods clearly, including hyperparameters. They have clean descriptions of each algorithm and easily readable text.

Nit: for table 2, it would be easier to read if top scores (or, top scores within stdev) were bolded.
Nit: for table 3 with 2 players and possibly for table 4, it might be better to report results using a graph as this will show the progress that each algorithm made and document where they were cut off. This could even just be in the appendix.
Nit: you use box-locking-easy and box-locking-simple interchangeably--best remove one.

## Originality
I’m not aware of other work which provides as clean a comparison of both different tasks and different methods, so there is novelty in the open source benchmark that this provides. The work contains some original tasks built on top of existing ones (i.e. in the Hide and Seek domain). There are no original algorithms in this paper as it is not the focus of the work.

## Significance
From the conclusion:
> “Given access to equally sized compute budgets for hyperparameter optimization, MAPPO is the most consistently successful algorithm across the studied environments, performing well at SMAC, as well as any algorithm in the MPEs and the best in hide-and-seek and Hanabi-small. Surprisingly, we also find that the centralized Q function variants DDPG, SAC, and TD3 almost entirely fail to solve any of the studied tasks, particularly in slightly harder environments.”

The main conclusion of the paper is “MAPPO is a good starting point for tackling new MARL problems”. This is a useful practical tip for researchers in the field. However, the work provides little insight into why that is the case. The next lines of the conclusion confirm that:

> “Of course, these results are likely dependent on the choice of hyperparameters and architectural choices. Thus it may be the case that the initial point around which we perform our grid search for MA-DDPG/TD3/SAC does not contain any good solutions and that performance of a set of hyperparameters on the MPEs is not predictive of performance in other domains. Finding a better set of hyper-parameters over which to search for the off policy algorithms is an important direction for future work.”

The authors admit that they were only able to explore a limited space of hyperparameters and their work may not hold in other circumstances. There is plenty of room for continued exploration of hyperparameters, certainly, but even more valuable would be any indicators into why different techniques might have different performance on the different tasks. I think in order for the publication to get accepted to a conference like ICLR it likely needs a bit more investigation, though researchers may certainly find the benchmark useful in research.

---

> ### Author Response · Authors · 2020-11-17
> **Thank you for the comments, here are a few responses**
>
> We thank the reviewer's feedback and suggestions. We have incorporated some of the stylistic suggestions into the updated version of our paper. A few points that we would like to address:
>
> >"The main conclusion of the paper is “MAPPO is a good starting point for tackling new MARL problems”. This is a useful practical tip for researchers in the field. However, the work provides little insight into why that is the case."
>
> The vast superiority of MAPPO compared to other off-policy algorithms is a surprising and interesting finding which is important due to the undervalued nature of MAPPO in MARL. We plan to investigate this further with additional ablations and experiments.
>
> >"for table 2, it would be easier to read if top scores (or, top scores within stdev) were bolded. for table 3 with 2 players and possibly for table 4, it might be better to report results using a graph as this will show the progress that each algorithm made and document where they were cut off. This could even just be in the appendix. Nit: you use box-locking-easy and box-locking-simple interchangeably--best remove one.""
>
> Thank you for the suggestions - we will update the paper to incorporate some of this feedback.

---

### Official Review · AnonReviewer2 · 2020-10-29
**Ok but not good enough**

**Rating:** 4
**Confidence:** 4

**Review:**

In order to compare different MARL algorithms more fairly, the author compared the differences in the performance of different algorithms(MADDPG, MATD3, MASAC, Qmix, MAPPO) in different environments(Particle-World, StarCraft Micromanagement, Hanabi, The Hide-and-Seek Domain).  The author's writing and organization are very good, so that I can clearly understand the content of the paper, and also has certain highlights, but I think it has not reached the ICLR criteria.

First of all, as an article about BENCHMARKING MULTI-AGENT DEEP REINFORCE- MENT LEARNING ALGORITHMS, I think the author should compare a series of algorithms to bring more insightful analysis and inspiration. Unfortunately, I found that the author simply I enumerate the performance differences of different algorithms in different environments, and there is also a lack of analysis of what methods are suitable for what tasks. I suggest that the author conduct further analysis and experiments, and also pay attention to the advantages and disadvantages of different algorithms. I hope that the author can adjust the content of the paper, and don't make people feel that the author has collected different official implementations, and then simply run them and compare directly.

In addition, I think the author may be missing some important MARL algorithms. For example, in terms of policy-base, it is recommended that the author consider adding algorithms such as COMA[1] and MAAC[2] (the authors of these algorithms have already announced their official implementations). At the same time, most of the author's algorithms have official implementations on the Internet. How to compare different algorithms fairly is of great concern, but the author's content is still relatively weak, and the author is recommended to improve this part of the discussion.

Overall, I vote for a rejection

[1] Counterfactual Multi-Agent Policy Gradients, Foerster et.al. AAAI 2018

[2] Actor-Attention-Critic for Multi-Agent Reinforcement Learning, Iqbal and Sha, ICML 2019

---

> ### Author Response · Authors · 2020-11-17
> **Thank you for the comments, here are a few responses**
>
> We appreciate the reviewers suggestions and insights, and address a few points below.
>
> > "For example, in terms of policy-base, it is recommended that the author consider adding algorithms such as COMA[1] and MAAC[2] (the authors of these algorithms have already announced their official implementations)."
>
> We did not benchmark with COMA due to the fact that MAPPO incorporates, and adds to, the core ideas in COMA - specifically, the idea of using a centralized value function, which we implement in MAPPO. MAAC demonstrates the utility of attention in multi-agent RL, and thus is not a separate algorithm per-say, which is why we did not include it on our benchmark.
>
> > Q2:That the author can adjust the content of the paper, and don't make people feel that the author has collected different official implementations, and then simply run them and compare directly.
>
> We thank the reviewer for this suggestion. However, we would like to emphasize that there is no existing unified, open-source implementation of various MARL algorithms, and no benchmarks or implementations in particular of MAPPO, MATD3, and MASAC. Furthermore, there are no benchmarks or implementations, with the exception of QMix, of recurrent variants of these algorithms, which we believe is a significant aspect of our results and work. Given the lack of benchmarks for MAPPO, its high performance is particularly noteworthy, which we believe to be an important aspect of this paper.
>
> > "I suggest that the author conduct further analysis and experiments, and also pay attention to the advantages and disadvantages of different algorithms.e that the author can adjust the content of the paper, and don't make people feel that the author has collected different official implementations, and then simply run them and compare directly."
>
> Thank you for the suggestion - we would like to point out that the results presented in our paper demonstrate the strength and robustness of MAPPO across domains, achieving comparable state of the art results in the Starcraft II domain and outperforming other popular algorithms in all other domains. This finding is significant to the community due to the fact that MAPPO is largely undervalued in MARL. To further investigate these findings, we plan to conduct further ablation studies. The development of a unified, well-test library of recurrent MARL algorithms all of which work on different types of environments, including the turn-based Hanabi environment, is additionally a contribution to the community which we believe is significant.

---

### Official Review · AnonReviewer1 · 2020-10-29

**Rating:** 3
**Confidence:** 4

**Review:**

The paper aims to benchmark a suite of Multi-Agent Deep Reinforcement Learning algorithms across different environments in the cooperative multi-agent setting. The paper compares standard algorithms alongside extensions of well-known policy gradient algorithms to the multi-agent setting, i.e. PPO (MAPPO), SAC (MASAC) and TD3 (MATD3). The paper investigates the ease of using an algorithm by doing a fair hyperparameter search for different algorithms. The paper concludes by saying that MAPPO is a promising choice for tackling a multi-agent problem.

Such a benchmarking paper would absolutely be useful. In its current form, however, it is not yet ready for publication. There are three primary reasons. First, the scope is misplaced: the paper starts with the scope of benchmarking different algorithms but seems to shift focus primarily to MAPPOs benefits. Second, the experiments only include 3 runs, with large overlapping deviations, limiting the ability to make claims about any significant differences. This is particularly problematic for a paper where the main focus is on experiments to provide insight. Finally, it seems like Pop-art is only used with MAPPO (please clarify this if I am mistaken), calling into question if the main reason for improvements in MAPPO is this choice. Reward scaling, done by Pop-art, can significantly affect learning performance as pointed out in Henderson et al. (2017). Using Pop-art only for MAPPO makes the comparison unfair between different approaches and potentially invalidates the conclusions presented in the paper.

As two other more minor points:

1. Multi-agent methods are built around numerous agents learning separate policies. The choice of sharing the same weights (Page 4, Sec 3.3) among different policies doesn’t seem to me like a fair choice. This also makes the algorithms different from their original intended use.

2. There are several inconsistencies or questionable choices in the paper:

a. As per appendix, MADDP, MATD3, MASAC, QMix have gamma parameters, whereas there is no such parameter mentioned for MAPPO. Having a different gamma value for the methods essentially changes the problem.

b. There is a difference between similar hyperparameters for different methods, i.e. MAPPO uses a gradient clipping of 20, whereas other methods use a gradient clipping of 10. How was this chosen and why is it different?

c. Appendix (Hanabi), MAPPO uses a learning rate of 7e-4, whereas the learning rate is not mentioned as a value being tested in the search grid.

d. MADDPG and MASAC are said to use a centralized critic, but according to Algorithm 1 and Algorithm 2, they seem to estimate separate critics.

e. MASAC should have a tuple of three parameters (value functions, q function and policy), but Algorithm 2 talks about only two parameters (q function and policy).


A few questions:
1. In SMAC environments, why are MASAC, MATD3 and MADDPG not run for other settings (Table 2)?

2. In MPE, I was not able to compare the results for MADDPG to the original paper. Can you point out the results (i.e. example the table number in Lowe et al. 2017)?  This also seems strange as Lowe et al. 2017 use a non-recurrent network with a different parameterization for each of the policies. In contrast, the current paper uses a recurrent network and shared policies, making the models quite different.

Suggestions:
1. Although the paper compares all the well-known policy gradient methods, comparing against a simple Vanilla Actor-Critic network would provide useful insight by removing some of the complexity.

2. It would be good to include hyperparameters of PPO like the \epsilon clipping.

3. The graphs in Figure 3 do not have a visible axis.

4. Including parameter sensitivity would provide much more insight into the properties of the algorithms,  and how they perform for general configuration of parameters.

5. Section 3.1, P(s’|s,u) : \mathcal{S} \times U \times \mathcal{S} \rightarrow [0,1], should have a U^n instead of U.

References
1. Henderson, P., Islam, R., Bachman, P., Pineau, J., Precup, D., & Meger, D. (2019). Deep Reinforcement Learning that Matters.
2. Lowe, R., Wu, Y., Tamar, A., Harb, J., Abbeel, P., & Mordatch, I. (2020). Multi-Agent Actor-Critic for Mixed Cooperative-Competitive Environments.

--- Update after reading the rebuttal and other reviews

Though some concerns have been addressed, a critical issue remains unresolved, which is that the experiments use only 3 runs.

As an additional point, it is useful that you've identified MAPPO as a good multi-agent algorithm. However, for clarity, it is better to focus the paper around MAPPO rather than strictly calling it a benchmarking paper. Further, it would be good to clearly state in the paper that the results now include reward normalization.

---

> ### Author Response · Authors · 2020-11-17
> **Thank you for the comments, here are a few responses**
>
> We thank the reviewer for some insightful comments and suggestions. We would like to respond to some of the highlighted issues, and clarify a few points:
>
> > "As per appendix, MADDPG, MATD3, MASAC, QMix have gamma parameters, whereas there is no such parameter mentioned for MAPPO. Having a different gamma value for the methods essentially changes the problem."
>
> Thank you for pointing this out - we use the same gamma discount factor (which is 0.99) for all experiments, and we have added this value for the MAPPO parameters in the appendix.
> > " There is a difference between similar hyperparameters for different methods, i.e. MAPPO uses a gradient clipping of 20, whereas other methods use a gradient clipping of 10. How was this chosen and why is it different?":
>
> We chose gradient clipping of 10 because this was the value used in the original QMix paper; to maintain consistency, we have rerun MAPPO experiments with a gradient clip value of 10 and have updated results, which remain largely unchanged.
>
> > "Appendix (Hanabi), MAPPO uses a learning rate of 7e-4, whereas the learning rate is not mentioned as a value being tested in the search grid."
>
> Thank you for pointing this out - we have added the learning rate search details in the appendix now.
>
> > "MADDPG and MASAC are said to use a centralized critic, but according to Algorithm 1 and Algorithm 2, they seem to estimate separate critics"
>
> These algorithms use centralized critics in that the critics take as input both the centralized/global state and all agents' actions. In a shared policy setting (which is the case for nearly all experiments), all agents share the same critic, but in an unshared policy setting, each agent will have its own associated critic (which still takes as input global information).
> > "MASAC should have a tuple of three parameters (value functions, q function and policy), but Algorithm 2 talks about only two parameters (q function and policy)."
>
> Our implementation of MASAC is built off of the SAC algorithm in Haarnoja Et. Al, 2016, which utilizes only policy and Q function networks.
> > "In SMAC environments, why are MASAC, MATD3 and MADDPG not run for other settings (Table 2)"
>
> In the previous version of our paper, due to the lack of success of MASAC, MATD3, and MADDPG in fairly easy tasks, we did not run them on harder maps. We now have run experiments with MASAC, MATD3, and MADDPG on all maps and have updated the results in these tables with these updated values - the trend continues to be that these algorithms have little success in these harder tasks.
> > "In MPE, I was not able to compare the results for MADDPG to the original paper. "
>
> The MADDPG paper (http://arxiv.org/abs/1706.02275) unfortunately does not include comprehensive results for all multiagent particle env tasks, such as simple_reference. Furthermore, they use normalized scores in the paper. - However, Figure 4 shows the training curve on speaker-listener, which we reproduce with our implementation.
>
> This also seems strange as Lowe et al. 2017 use a non-recurrent network with a different parameterization for each of the policies. In contrast, the current paper uses a recurrent network and shared policies, making the models quite different.
>
> It is true that the original MADDPG algorithms were implemented with MLP networks and unshared policies; however, we use recurrent networks for two reasons: first, all environments are partially observed with the exception of the MPE environment - thus, algorithms benefit from using recurrent networks, as they are able to carry forward observation history. Second, the original implementation of QMix utilizes recurrent networks - hence, for consistency across algorithms, each algorithm utilizes recurrent networks. We utilize shared policies for the same reason - in all environments except for the speaker-listener MPE environment, agents are homogenous, allowing for reasonable use of shared policies. Furthermore, the original QMix paper shares policies across agents, so for consistency, we use shared policies for all algorithms when possible.
>
> > "The scope is misplaced: the paper starts with the scope of benchmarking different algorithms but seems to shift focus primarily to MAPPOs benefits.
>
> We benchmark each algorithm by giving each a hyperparameter search space of equal size, and evaluating their performance on each testing environment. We ultimately find that MAPPO is the strongest, and due to the clear success of MAPPO compared to the other algorithms, we emphasize it's strengths, particularly because the promise and benefits of multi-agent PPO are largely ignored in the literature.
> > "It seems like Pop-art is only used with MAPPO (please clarify this if I am mistaken), calling into question if the main reason for improvements in MAPPO is this choice. "
>
> We have added reward-normalization to our off policy algorithm results , which do not affect our results significantly.

---

### Official Review · AnonReviewer3 · 2020-10-29
**Comparison of MARL algorithms that could benefit the community**

**Rating:** 6
**Confidence:** 3

**Review:**

The major contribution of this paper is benchmarking 5 MARL algorithms on 4 cooperative multi-agent environments. Also, this paper found that under constrained hyperparameter search budgets, the multi-agent PPO algorithm has more consistent performance over the other algorithms across different tested multi-agent environments. The code base is open-sourced for public use, which benefits the MARL community.

Researchers in MARL often find it difficult to find a useful benchmark for multiagent learning algorithms and multiagent environments. Thus this paper makes a good contribution to the community, but, on the other hand, since this paper is not really intended to present any major technical contributions, that could be considered a weak point for an ICLR submission.

To the best of the reviewer's knowledge, there has not been any previous work that tries to produce a benchmark for multi-agent deep reinforcement learning. As the first work that attempts to fill this gap, this paper presents a comprehensive implementation of popular MARL algorithms tested on a representative list of MA environments, which is the major reason for acceptance suggestion. But there are some limitations of this paper. For example, there is little discussion about the results and the authors did not attempt to do more technical investigation for understanding the their findings about the algorithms. In this sense, this paper is less than a qualified research paper.
In particular, it would be helpful if  the authors have any insight on why the MAPPO works more consistently than the other algorithms.

One major suggestion is to add standard deviation to the numeric values presented in Table 1, 2, 3 and 4.

If this paper get accepted to this conference, peer researchers could be incentivized to contribute similarly comprehensive benchmark to this field, which would benefit the advancements of this field in the long run.

I have read over the rebuttal and discussion and will keep my evaluation score unchanged as I see value in benchmarking papers such as this for the community.

---

> ### Author Response · Authors · 2020-11-17
> **Thank you for the comments, here are a few responses**
>
> We thank the reviewer for their feedback, and for the positive comments regarding our work. We’d like to respond to a few comments:
>
> > "there is little discussion about the results and the authors did not attempt to do more technical investigation for understanding their findings about the algorithms. "
>
> The main contribution we present in this paper is that we demonstrate the robustness and strength of MAPPO on every domain, and it's superior performance compared to other algorithms. We do hope to investigate the reasons for these findings through further experiments and ablations, but we believe our findings are important to the community due to the fact that MAPPO is largely undervalued in MARL. We hope to investigate these findings more through additional experiments and ablations. We'd also like to emphasize that an open-source, unified library of recurrent MARL algorithms, compatible with different types of environments, including Hanabi and Starcraft II, is an important contribution to the community.
>
> > "One major suggestion is to add standard deviation to the numeric values presented in Table 1, 2, 3 and 4."
>
> Thank you for the suggestion - we will look into adding this into our tables.

---

### Author Response · Authors · 2020-11-17
**Paper updated with changes in red**

We have updated our paper to incorporate all the feedback from reviewers. All the changes are colored red.

---

### Decision · Program_Chairs · 2021-01-07
**Final Decision**

**Decision:**

Reject

**Comment:**

The paper presents a thorough comparison of different algorithms for multi-agent Deep RL methods. The conclusions of the paper is that across a variety of envionment and hyperparameter tuning, multi-agent PPO seems to peform well relatively to the competitors.

The reviewers agreed that the paper fills a gap in the literature regarding a fair and thorough comparison of algorithm, and that the paper clearly presents the results. As it stands, the code to reproduce the experiments and the results are a useful contribution to the community. However, the reviewers felt the technical contribution to be below the bar for ICLR, as the paper does not help in understanding the differences between algorithms, or develop insights as to how to further improve algorithms. The large standard deviations of the various algorithms also makes the main experimental insight (MAPPO works consistently well) relatively weak.